# TutorBench: A Benchmark To Assess Tutoring Capabilities Of Large Language Models

## Abstract

As students increasingly adopt large language models (LLMs) as learning aids, it is crucial to build models that are adept at handling the nuances of tutoring: they need to identify the core needs of students, be adaptive, provide personalized guidance, and be accurate. To this end, we introduce TutorBench, a dataset and evaluation benchmark designed to rigorously evaluate the *core tutoring skills* of LLMs. The dataset comprises 1,490 samples curated by human experts, focused on high-school and AP-level curricula. The samples are drawn from three common tutoring tasks: (i) generating adaptive explanations tailored to a student's confusion, (ii) providing actionable feedback on a student's work, and (iii) promoting active learning through effective hint generation. To account for the inherent complexity of tutoring, samples are accompanied by sample-specific rubrics which are used to judge model responses during evaluation. TutorBench uses a reliable and fine-grained automatic evaluation method that uses an LLM-judge and the sample-specific rubrics. We evaluate 16 frontier LLMs on TutorBench and present a detailed analysis of their performance and behavior. Our results show that none of the frontier LLMs achieve a score of greater than 56%, showing a large room for improvement. We find that LLMs fall short in exhibiting the full range of tutoring skills needed to guide, diagnose, and support students effectively, with all the frontier models achieving less than a 60% pass rate on rubric criteria related to these skills. We also find that different model families exhibit varied strengths and limitations: the Claude models outperform others in supporting active learning, while they lag behind in the other two use cases. By releasing TutorBench, we provide a comprehensive and unsaturated benchmark to guide the development of the next-generation of AI tutors.

## 1 Introduction

Large language models (LLMs) are rapidly transforming the way students learn and access educational support (Handa et al., 2025; Ammari et al., 2025; Geraghty & Goldstein, 2024; Scarlatos et al., 2025). Tools like ChatGPT, Gemini, and Claude have already serve as on-demand tutors for millions of learners worldwide. To encourage adoption, many providers offer variants specializing in tutoring (OpenAI, 2025b) and extend free access to students (Google, 2025). It is clear tutoring stands as a priority for model builders because it holds the promise of delivering transformative, personalized education at low cost. However, the true impact of AI-based tutoring is not yet fully understood: while many studies show improvements in learning outcomes (Vanzo et al., 2025), there are also reports that students perform worse when the tools are taken away (Bastani et al., 2025).

Given their rapid adoption, it is crucial to study and benchmark LLM behavior when they serve as tutors. Many existing LLM benchmarks focus on evaluating LLMs on advanced domain knowledge and reasoning (Phan et al., 2025; Glazer et al., 2024; He et al., 2024). While these dimensions are crucial for understanding the limits of model performance, they overlook the more nuanced, human-centered capabilities necessary for tutoring, such as clear and thorough explanation, adaptivity to a learner's needs, and providing the right guidance. Some recent efforts have begun to explore how to benchmark LLMs in tutoring contexts (Maurya et al., 2025; Gupta et al., 2025). However, they are limited in scope as they focus on text-only examples from just one or two subjects like mathematics or computer science. This differs significantly from real-world usage, where students frequently upload pictures of their work and seek help on several subjects. SOTA LLMs such as GPT-4 achieve nearly perfect scores on both of these benchmarks according to their own reported results, indicating

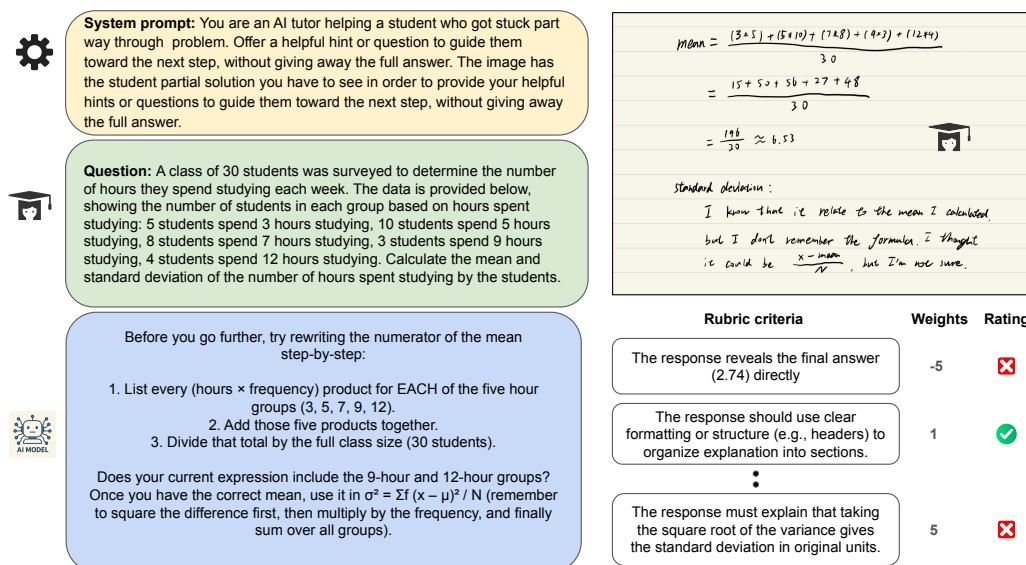

Figure 1: An example from the TUTORBENCH dataset. Each sample includes a **system prompt** defining the tutoring goal (top left), a **question** along with the student's partial work (top right), an **AI model's response** (bottom left), and a set of **rubric criteria** for evaluation (bottom right). In this instance, the AI is prompted to provide a hint for a statistics problem.

the need for more nuanced benchmarks that expose shortcomings in model behavior. Moreover, they lack reliable auto-judge methods to evaluate LLMs' tutoring capabilities.

To address the above gaps, we present **TUTORBENCH**[1], a benchmark to evaluate the tutoring capabilities of LLMs. TUTORBENCH consists of 1490 conversations between a student-persona and a tutor-persona. The conversations cover 6 STEM subjects (Biology, Physics, Chemistry, Statistics, Calculus, and Computer Science) and focus on high school and AP curricula. The conversations are designed to reflect real-world usage by students and focus on three tutoring use cases: adaptive explanation generation, feedback and assessment, and active learning support. These use cases were chosen to test an LLM's ability to calibrate its responses to individual students and support their learning journey rather than simply generating universal and standard solutions to questions. We provide more details about the use cases in Section 2.1. TUTORBENCH is also **multimodal**, with both text-only and image-based conversations. 828 samples in the dataset contain images of handwritten or printed work by students, reflecting real-world usage by students.

We design TUTORBENCH to be a **rubrics-based** evaluation to account for the open-ended nature of tutoring. Each sample in the dataset is accompanied by a set of sample-specific evaluation rubrics written by human experts that are self-contained, mutually exclusive, and verifiable. Together, these rubrics capture the requirements of a desirable response to the corresponding test sample. Model responses are graded by an LLM judge with a pass/fail rating on each rubric criterion.

We conduct data collection of TUTORBENCH with the help of human experts. The questions and the rubrics to each question were written by experts of the corresponding subject who hold a Bachelor's or higher degree and have either tutoring or professional experience in the corresponding subject. In order to guarantee the difficulty level of TUTORBENCH, we then prompt 5 state-of-the-art LLMs (Gemini 2.5 Pro Google DeepMind (2025), Claude 3.7 Sonnet Anthropic (2025a), o3 OpenAI (2025a), DeepSeek-R1 DeepSeek (2025) and Llama 4 Maverick Meta AI (2025)) to respond to each collected test sample. We retain only conversations that result in a score of less than 50% for at least three of the five models. This results in a challenging benchmark, with the best performing model attaining a score of 55.65%.

---

[1]A sample subset of TUTORBENCH can be found at https://huggingface.co/datasets/tutorbench/tutorbench. The full dataset will be released soon.

Using the samples in TUTORBENCH, we evaluate and analyze 16 frontier LLMs. We report the overall scores of the models in Section 3.1. Among all the models tested, Gemini 2.5 Pro and GPT-5 achieve the best performance, with $55.65\%$ and $55.33\%$ final scores respectively ($100\%$ being the maximum final score). The Claude model series falls behind the Gemini 2.5 Pro and GPT model series with non-negligible performance gaps, with Claude Opus 4.1 (Thinking) achieving a final score of $50.78\%$. We observe that models achieve a score of only $47.16\%$ on the use case of *adaptive explanation generation*, indicating that frontier LLMs still struggle to generate effective personalized responses. Further, they also achieve less than $55\%$ on the other two use cases. Interestingly, models of the Claude series outperform others in the "*active learning support*" use case, while trailing in the overall performance. More detailed analysis and observations with respect to the use cases and evaluation dimensions can be found in Section 3.2 and Section 3.3.

## 2 BENCHMARK DESIGN

We design TUTORBENCH to cover a broad spectrum of common, real-world tutoring use cases, including both text-only and multimodal questions on 6 STEM subjects. Use cases in TUTOR-BENCH focus on assessing human-centered capabilities necessary for tutoring, such as explanation, guidance, and adaptivity to a learner's needs. We also design a reliable rubric-based eval to automatically assess LLM capabilities on such subjective matters. Details regarding the data collection process are provided in the Appendix (Section A.1).

### 2.1 TUTORING USE CASES

As large language models (LLMs) become increasingly integrated into educational workflows, it is important to rigorously evaluate their effectiveness in different tutoring scenarios. TUTORBENCH captures three core use cases that reflect essential tutoring behaviors: adaptive explanation, assessment and feedback, and active learning support. We elaborate on these 3 use cases below.

**Use Case 1: Adaptive Explanation Generation.** Personalized instructions can help unlock a student's potential. One of the most impactful qualities of a good human tutor is their ability to adapt explanations based on a student's current understanding and knowledge gaps. The key skill required here is to identify core misconceptions and shortcomings in a student's understanding and address them in a way that is easy to understand.

To evaluate this capability, we design a multi-turn interaction setup consisting of: an initial question written in a *student-persona*, an initial explanation written in a *tutor-persona*, and a follow-up question written in a student-persona. During evaluation, an LLM is presented with this triplet, along with a system prompt/instruction to generate an explanation that directly addresses the knowledge gap or misunderstanding exhibited in the follow-up question by the student. An example conversation is shown in Fig. 2. Model responses are evaluated on whether they can recognize the specific context and background implied by the student's follow-up, and whether they can produce a helpful, focused, and to-the-point response tailored to the student's current understanding. These aspects are captured in the set of sample-specific rubric criteria that accompany every sample in the dataset.

**Use Case 2: Assessment and Feedback.** Students (or supervisors) often use LLMs as a means to assess their (or their students') work and get instant feedback. This reflects one of the most promising applications of LLMs: providing real-time evaluation and correction that helps learners iterate quickly. To capture this scenario, TUTORBENCH includes examples where the model is shown a student solution and is asked to assess it. The model is expected to analyze the work, identify any mistakes, classify the nature of each error (e.g., arithmetic, factual, conceptual, etc), and generate feedback. An example of this use case is shown in Fig. 2.

**Use Case 3: Active Learning Support.** An essential aspect of high-quality tutoring is to promote active learning, wherein students are encouraged to engage directly with the problem-solving process rather than passively receiving answers. Effective tutors guide students toward the solution through scaffolding, such as hints, analogies, or intermediate steps, which support learning while preserving student agency.

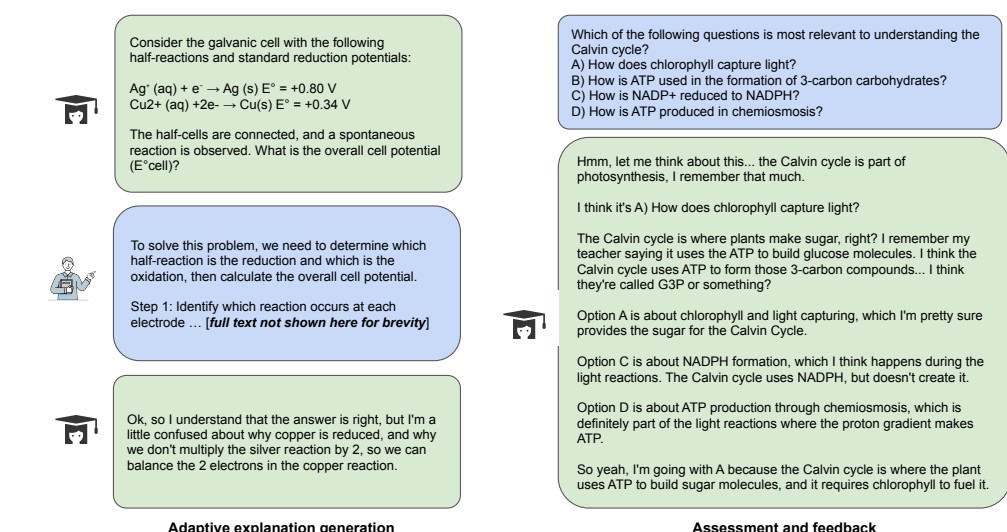

Figure 2: Examples of two core use cases for an AI tutor. **Left:** The *adaptive explanation generation* scenario, showcasing a multi-turn dialogue in electrochemistry. The system must provide a targeted clarification in response to a student's specific follow-up question regarding standard reduction potentials. **Right:** The *assessment and feedback* scenario, where a student provides a reasoned but incorrect answer to a biology question. The system's task is to analyze the student's reasoning, identify the misconception about the Calvin cycle, and provide corrective feedback.

To test this behavior, we present LLMs with incorrect or partially correct student responses and ask them to generate helpful hints that enable the student to take the next step. These hints must strike a careful balance, providing just enough information to resolve confusion without directly giving away the final answer. This task simulates scenarios where a student is stuck midway and requires targeted intervention to proceed. An example of this use case is presented in Fig. 1.

## 2.2 MULTIMODALITY

One key feature of TUTORBENCH is its multimodal design, reflecting the authentic ways students engage with LLMs for tutoring. For each use case illustrated above, we create both text-only and multimodal samples. The images in the dataset consist of hand-written text/diagrams, typed content, or screenshots, mirroring real settings where students often communicate not only through written text but also by sharing images of their work or problem-solving steps. This multimodal component is essential for evaluating LLMs as effective AI tutors, since it tests their ability to interpret and provide feedback across different forms of student input. An example of a multimodal sample is shown in Fig. 1 where the student solution is in the form of an image of their hand-written work.

## 2.3 LLM JUDGE WITH SAMPLE-SPECIFIC RUBRICS

Given the open-ended nature of the three use cases described above, we adopt a rubric-based evaluation using an LLM-judge based on Claude Sonnet 4 (Anthropic, 2025b) to evaluate model responses. Similar to recent research on rubrics-based model training and evaluation Arora et al. (2025), TU-TORBENCH creates separate rubric criteria for each example (ranging from 3 to 39 per example) that help capture fine-grained qualities of a given response. This results in a total of 15,220 rubrics across the whole dataset. These criteria enable efficient and reliable evaluation using an LLM-judge, bypassing a time-consuming human judge for every new model release.

Rubric criteria are designed to be self-contained, mutually exclusive, and collectively comprehensive. This design ensures that they can be applied consistently and without overlap, while also covering the full space of desirable response qualities. Granularity of the criteria makes them partic-

ularly well-suited for grading with LLM-based judges. An illustration of the sample-specific rubrics is shown in Fig. 1 (bottom right). More examples can be found in the sample dataset, a URL to which is provided above.

We additionally introduce a weighting scheme to reflect the relative importance of different criteria. Rubrics designated as *critical* are assigned a weight of 5, while *non-critical* rubrics carry a weight of 1. In some cases, *critical* rubrics may be assigned a negative weight of -5 to penalize undesirable behaviors. For instance, in active learning settings, a rubric may require that the final answer not be revealed directly, and a violation of this principle would result in a strong negative score. To quantify overall model performance, we compute a weighted average of the binary scores assigned to each criterion. The final score is normalized to the range [0,1].

We conduct thorough experiments to show the alignment of the LLM-judge vs the human-expert judge. To study the quality of the LLM-judge and compare it with human experts, we collect 3 human ratings per rubric criterion on a subset of 250 samples. Our experiments show that the LLM-judge aligns better with the human experts than the median human expert, and achieves an F1 score of 0.81 with respect to the majority vote. Details of experiments are described in Section 3.7.

## 2.4 RUBRIC CRITERIA TAGS

To further enable fine-grained analysis of model performance, we annotate each rubric criterion with four tags. Firstly, we annotate each rubric with one of the following *evaluation dimensions* to capture the primary axis along which the model is being judged: instruction following, style and tone, truthfulness, visual reasoning, visual perception, conciseness and relevance, student level calibration, and emotional component. These dimensions provide a structured lens for understanding different facets of tutoring quality. We provide definitions of these dimensions in Appendix A.3. Secondly, we tag the rubric criteria with specific *tutoring skills* being assessed: asking guiding questions, identifying core misconceptions, recognizing correct or incorrect student steps, including examples or analogies, providing alternative solutions, stating definitions or theorems (or other knowledge), or providing step-by-step help. Together, these two tags allow us to analyze not just whether the model performs well, but also what type of tutoring capability is being demonstrated or lacking.

In addition, we annotate each rubric criterion with two complementary tags that enrich the dataset. We annotate each rubric as being *explicit or implicit*, to indicate whether the criterion is related to an explicit request in the prompt or implicitly inferred from the tutoring context. We also annotate criteria as *objective or subjective* to denote whether the criterion judges a response according to an objective standard (e.g., factual accuracy, correctness of a step) or a more subjective one (e.g., appropriateness of tone, empathy). We provide the distributions of the various tags in Appendix A.5.

## 3 EVALUATION AND ANALYSIS

### 3.1 OVERALL MODEL PERFORMANCE

Model performance on TUTORBENCH is evaluated using the weighted average rubric rating $\mathrm{ARR}_w$, defined on the $j$-th example as

$$\mathrm{ARR}_w^j = \frac{\sum_{i=1}^{N_j} w_i^j \cdot \mathbb{1}_{r_i^j}}{\sum_{i=1}^{N_j} w_i^j \cdot \mathbb{1}_{w_i^j > 0}}, \tag{1}$$

where $N_j$ is the number of criteria in the $j$-th example, $w_i^j \in \{-5, 1, 5\}$ is the weight of the $i$-th rubric for the $j$-th sample, and $r_i^j \in 0, 1$ is the fail/pass rating of the model response on the $i$-th rubric of the $j$-th sample. The final score for a model is the average $\mathrm{ARR}_w$ across all the examples.

The evaluation results are presented in Table 1. We observe that Gemini 2.5 Pro (Google Deep-Mind, 2025) and GPT-5 (GPT-5, 2025) obtain the best overall performance, with very similar scores on both text-only and multimodal tests. Furthermore, none of the models surpass 56% overall performance, highlighting the complex nature of TUTORBENCH. Finally, it is noteworthy that the recently released gpt-oss-120b model performs close to the best models on the text-only subset.

| Rank | Model | Text-Only (%) | Multimodal (%) | Overall (%) | CI |
|------|-------|---------------|----------------|-------------|-----|
| 1 | Gemini 2.5 Pro | **57.05** | **54.53** | **55.65** | ± 1.11 |
| 2 | GPT-5 | **57.03** | 53.97 | 55.33 | ± 1.02 |
| 3 | o3 Pro | 56.07 | 53.45 | 54.62 | ± 1.02 |
| 4 | o3 Medium Effort | 54.11 | 51.68 | 52.76 | ± 1.00 |
| 5 | o3 High Effort | 52.91 | 51.43 | 52.09 | ± 1.01 |
| 6 | Claude Opus 4.1 (Thinking) | 51.65 | 50.08 | 50.78 | ± 1.05 |
| 7 | Claude Opus 4 (Thinking) | 50.40 | 49.14 | 49.71 | ± 1.02 |
| 8 | Claude Opus 4.1 | 49.51 | 45.72 | 47.40 | ± 1.06 |
| 9 | Claude 3.7 Sonnet (Thinking) | 45.67 | 47.07 | 46.45 | ± 1.03 |
| 10 | Claude Opus 4 | 47.79 | 43.59 | 45.46 | ± 1.06 |
| 11 | Llama 4 Maverick | 39.54 | 40.73 | 40.20 | ± 1.00 |
| 12 | GPT-4o | 39.10 | 33.74 | 36.12 | ± 0.96 |
| | gpt-oss-120b | 56.01 | N/A | N/A | ± 1.49 |
| | gpt-oss-20b | 49.01 | N/A | N/A | ± 1.53 |
| | DeepSeek-R1 | 48.38 | N/A | N/A | ± 1.50 |

Table 1: Evaluation of state-of-the-art LLMs on TUTORBENCH. Gemini 2.5 Pro achieves the highest score (55.65%), highlighting the complex nature of the samples in TUTORBENCH and the need for further advancement in models for tutoring applications. We report scores on the text-only samples for gpt-oss-120b, gpt-oss-20b, and DeepSeek-R1 because they do not support multimodal input.

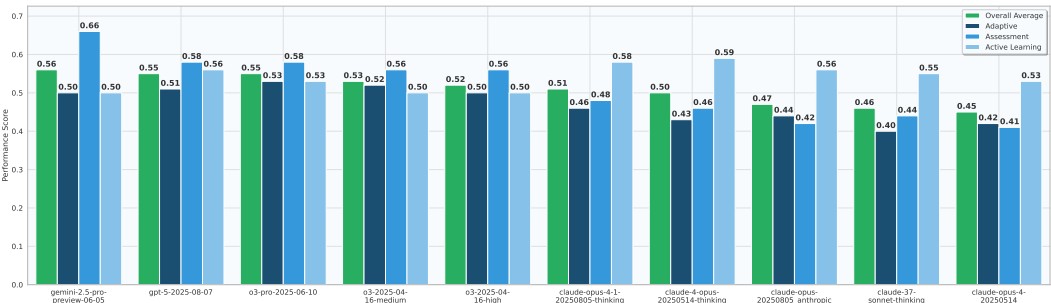

Figure 3: Model performance across three use cases: **Adaptive**, **Assessment**, and **Active Learning**. We observe a distinct difference between the performance of the Claude family of models compared to the other models, with the Claude models performing significantly better in providing active learning support, but still lagging behind other models overall.

## 3.2 PERFORMANCE BY USE CASE

Performance breakdown by use case is shown in Fig. 3. On average, models achieve a score of 47.16% on adaptive explanation generation, 51.56% on assessment and feedback, and 54.07% on active learning support. We also observe that the Claude family of models performs much better in the active learning support use case compared to the other models, although their overall performance is poorer.

## 3.3 PERFORMANCE ALONG EVALUATION DIMENSIONS

As explained in Section 2.4, we annotate each rubric criterion with one **evaluation dimensions**. This allows us to aggregate the pass/fail ratings on subsets with each of the tags. We present the mean pass/fail rate along these dimensions for the top 10 models (according to Table 1) in Fig. 4. While performance along these dimensions roughly follows the same trend as the overall performance, Gemini 2.5 Pro shows a much better performance than other models in recognizing student emotions such as confusion, frustration, curiosity, and generates responses with the right tone and style (e.g., by using headings, bullets, and LaTeX). On the other hand, GPT-5 and o3 Pro perform best on factual correctness (truthfulness), student-level calibration, and instruction following.

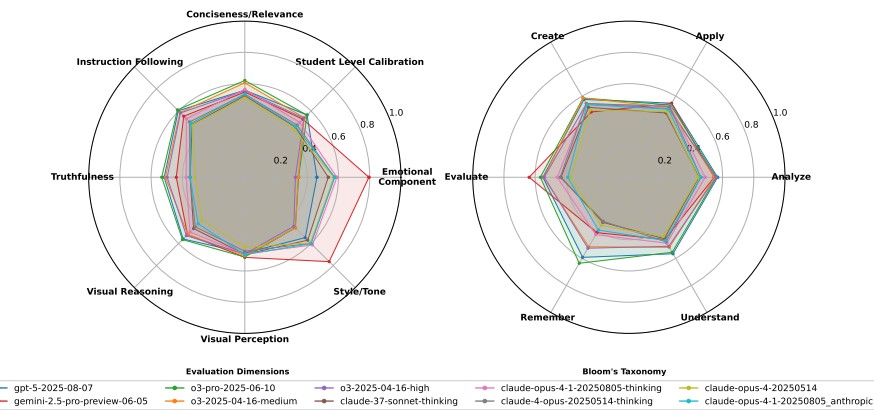

Figure 4: Model performance breakdown along evaluation dimensions and Bloom's taxonomy categories. While the top-performing models GPT-5 and Gemini 2.5 Pro are close overall, their performance differs widely when measured along the above dimensions.

## 3.4 PERFORMANCE ALONG DIFFICULTY LEVEL

This study categorizes each sample using Bloom's taxonomy, which organizes cognitive tasks into skill levels ranging from remembering and understanding to advanced ones such as analyzing, evaluating, and creating, in order to better assess model performance across different levels of cognitive demand. Three state-of-the-art LLMs (Gemini 2.5 Pro, Claude Sonnet 4, and GPT-5) were used to annotate each sample, with a category label assigned if at least two models agreed, covering over 97% of the samples. Final scores for each category were calculated by averaging $ARR_w$ across all annotated samples. Results, shown in Figure 4, reveal that performance does not align with the taxonomy's order of difficulty. For example, Gemini 2.5 Pro scores lower on "remember" than on "evaluate", suggesting that while models may demonstrate advanced reasoning, they can struggle with recalling and explicitly stating content (when not explicitly prompted to), which is crucial for tutoring tasks that require effective context-driven communication.

## 3.5 PERFORMANCE ALONG TUTORING SKILLS

Tutoring is a complex task requiring several nuanced skills, such as calibrating to the student, recalling and applying knowledge, or breaking down problems into simpler steps. To evaluate models along with such finer skills, we classify the rubric criteria written by human-experts into 8 high-level skills: identifying core difficulty, identifying students' correct steps, identifying students' incorrect steps, recalling and stating knowledge, providing alternative solutions, including examples, asking questions to guide students, and providing step-by-step help. These skills were identified by manually inspecting 600 rubric criteria and by using an LLM to tag all the rubric criteria with one of the skills, if applicable.

We present the average pass/fail rate of LLM responses on these tags in Figure 5. Overall, we observe that models perform their best in identifying correct/incorrect steps by students (average scores of 53.7% and 53%), but they struggle to include alternate solutions, and examples/analogies in their responses (achieving average scores of 41.9% and 32.8%). Model performance varies across model families, with Gemini 2.5 Pro excelling at identifying correct/ incorrect steps and providing examples, while GPT-5 is stronger at spotting errors and misconceptions.

## 3.6 AN EVALUATION OF GPT-5 STUDY MODE

Along with the models in Table 1, we also evaluated OpenAI's recently released *study mode* OpenAI (2025b). According to their website, study mode is "powered by custom system instructions written in collaboration with teachers, scientists, and pedagogy experts" to promote deeper learning. Since it is only available via the web interface, we collected responses from the website.

Unlike API models, study mode does not allow direct system prompts and instead engages users in back-and-forth dialogue calibrated to their skill level. This makes apples-to-apples comparison with TUTORBENCH difficult, since TUTORBENCH evaluates only the final response in a conversation with a pre-specified history. For this reason, we exclude study mode from the main leaderboard and report results separately.

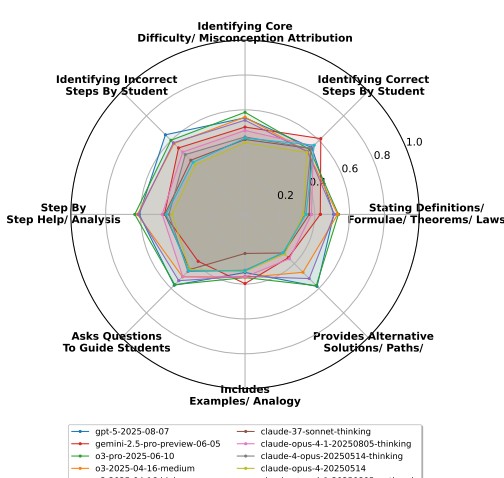

Figure 5: Model performance breakdown along tutoring skills: models struggle to include alternative solutions, examples, and analogies in their responses. However, they perform relatively better in identifying mistakes, correct steps, and core misconceptions.

On TUTORBENCH, study mode scores $46.94 \pm 1.06\%$ overall ($48.19\%$ text-only, $45.93\%$ multimodal), indicating poor performance against rubric criteria. Comparing with API-based GPT-5, we note two key observations: (1) study mode often provides only partial, intermediate responses, typically ending with a counterquestion (e.g., "Before we move on, do you want me to walk you through ...?"). These incomplete answers naturally score lower. (2) Even when complete, study mode responses perform worse than GPT-5's, as they are more concise and address fewer rubric criteria.

Thus, study mode's lower scores stem from both TUTORBENCH 's unsuitability for partial responses and from study mode's tendency to neglect several rubric criteria even in complete answers. We provide examples and more details in the appendix (Section A.4).

## 3.7 QUALITY OF LLM-JUDGE USED IN AUTOMATED EVALUATIONS

To achieve scalable evaluation, we rely on an LLM-judge to grade model responses against the rubric criteria. We experimented with multiple state-of-the-art models and chose Claude-4-Sonnet as our judge model, as it achieved the best alignment with human ratings.

To compare the LLM-judge with a human expert, we first collect 3 separate human-expert ratings across 250 examples, with a total of 2475 rubric criteria. To capture the inter-human ratings agreement, we select all the rubric criteria rated by each human-expert, and compare them to the other two ratings. The mean interhuman agreement across the 69 experts was 0.75, while agreement between the LLM-judge and human ratings was 0.78. The distribution of agreements per individual human expert who contributed to our annotations is shown in Fig. 6, along with the LLM-judge agreement.

We also study the alignment of the LLM-judge with the majority vote on rubrics. Firstly, we filter out the *critical* rubric criteria (with weights +5 or -5) from the 250 examples, resulting in a total of 1900 criteria with 3 ratings each. We then use the majority vote (pass/fail) as the label for each rubric. The LLM judge

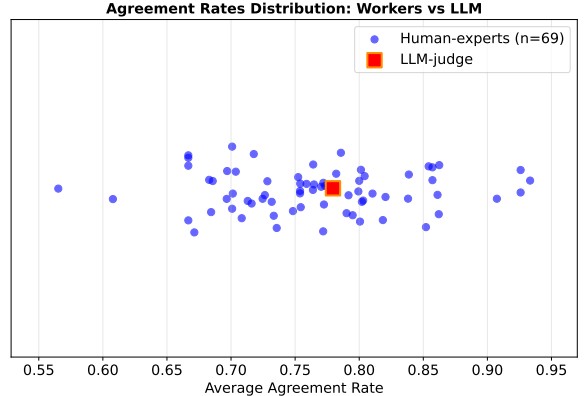

Figure 6: We measure the quality of the LLM-judge used in our evaluations by comparing its rating on model responses across 250 samples from TUTORBENCH with the majority vote obtained using 3 human-expert ratings. We observe that the LLM-judge ranks better than the median human-expert, thus demonstrating a strong alignment of the LLM-judge with human ratings.

achieves an F1-score of 0.82. The highest F1-score achieved by a single human-expert was 0.91, computed across a set of 321 criteria that were rated by them. This demonstrates that the LLM-judge has a strong performance and aligns well with human-expert ratings.

## 4 RELATED WORK

**Evaluation Frameworks for AI-Tutor Models.**  Evaluating tutoring quality has emerged as an important area of study. Maurya et al. (2025) propose MRBench, an evaluation benchmark with 192 math tutoring conversations. They propose a uniform taxonomy for evaluating tutoring capabilities based on learning principles. In a similar effort to evaluate the impact of using LLM-based tutors, Lyu et al. (2024) conduct a field study of 50 students by using an LLM-based coding assistant. While the students who used the assistant achieved statistically significant improvement in their performance, they expressed concerns about the limited role the assistant played in developing critical thinking skills. Similarly, Liu et al. (2024) document the integration of AI-based tools into Harvard's CS50 course.

**LLMs as Tutors.**  Recent research has also explored designing improved tutors using LLMs. Pal Chowdhury et al. (2024) develop a structured, rule-based tutoring framework with LLMs. Their system embeds guardrails and predefined pedagogical strategies, allowing LLMs to generate tutor responses while maintaining instructional control. Compared to free-form GPT-4 tutors, this hybrid approach reduced redundant or unhelpful dialog turns and better adhered to pedagogical principles. Similarly, Wang et al. (2024) propose Bridge, which uses cognitive task analysis to capture expert teachers' decision-making processes, such as diagnosing student errors and selecting remediation strategies, before feeding them into LLMs.

In contrast to these efforts, our dataset targets multimodal tutoring across six high school STEM subjects, moving beyond single-domain math word problems. Our work also introduces explicit rubric-based evaluation criteria tied to each sample, enabling efficient and scalable evaluation of multiple models. Our dataset is also substantially larger and covers a diverse set of use cases, topics, and evaluation dimensions. Overall, TUTORBENCH provides a broader testbed for measuring and advancing LLMs' tutoring capabilities.

## 5 LIMITATIONS OF TUTORBENCH

While our dataset provides a structured foundation for evaluating LLM tutoring, it has important limitations. Its scope of instructional use cases is constrained: we focus on three representative tutoring scenarios but omit other valuable tasks such as generating practice problems, designing exercises, or introducing new concepts. The dataset also evaluates only final responses to pre-formulated conversations, limiting assessment of adaptability in dynamic, multi-turn exchanges. Our dataset only incorporates images of students' work as input and does not test visual content generation. Our STEM focus provides depth in reasoning and problem-solving but excludes humanities domains that rely on narrative and interpretive skills. Finally, we isolate model outputs from the broader context of UI/UX design.

## 6 CONCLUSION

We introduce TUTORBENCH, a dataset and evaluation benchmark to assess tutoring capabilities of LLMs. We focus on three specific user scenarios: (i) adapting explanations to a student with a given background, (ii) providing assessment of students' work, and (iii) providing hints for active learning. TUTORBENCH is a rubrics-based evaluation and each sample in the dataset is accompanied by specific rubric criteria. These rubric criteria are weighted based on importance and are also tagged with several attributes that allow for finer analysis. Another important aspect of TUTORBENCH is the support for multimodal input: the dataset includes samples with images containing students' work to reflect real-world usage. We also provide a comprehensive evaluation of a select set of state-of-the-art LLMs and identify growth areas. Overall, TUTORBENCH serves as one of the first datasets to comprehensively evaluate LLMs on tutoring-based applications.

**Reproducibility Statement**   To facilitate reproducibility of our results, we provide a sample version of our dataset, which is publicly available at `https://huggingface.co/datasets/tutorbench/tutorbench`. The sample dataset consists of 30 samples, 10 from each use-case (with 5 text-only and 5 multimodal). We will soon release the full dataset along with the evaluation code to enable exact reproduction of our experiments and to support future research on improving the tutoring capabilities of large language models.

**LLM usage acknowledgement:**   We acknowledge that large language models (LLMs) were used to assist in the preparation of this manuscript, specifically for improving clarity and polishing the language of certain sentences. All substantive ideas, analyses, and conclusions remain the responsibility of the authors.

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

# A APPENDIX

## A.1 DATA COLLECTION PROCESS

TUTORBENCH consists of high-school-level questions from STEM subjects. The questions and the rubrics to each question were written by experts of the corresponding subject who have a Bachelor's or higher degree and have either tutoring or professional experience in the corresponding subject. Each example starts with a question, which is then followed by use-case-specific content. For the adaptive explanation use case, it is followed by (i) a teacher's explanation of the answer and (ii) a student's follow-up question asking for clarification on a specific part of that explanation. For the feedback and assessment use case, the initial question is followed by an incorrect solution written in a student persona. For the active learning support use case, the initial question is followed by a partial solution written in a student persona.

Subsequently, the human experts write a list of rubric criteria to evaluate model responses. The rubric criteria are formulated based on a "golden tutoring response" written by them. Each rubric criterion is annotated with several tags as explained in Section 2.4, and assigned a weight to indicate their importance. The weighting system is discussed in Section 2.3.

Five state-of-the-art models, Gemini 2.5 Pro, GPT-o3, Claude 3.7 Sonnet , DeepSeek-R1, Llama 4 Maverick are then prompted with the full example, along with use-case-specific instructions. The obtained responses are then graded against the rubric criteria. We keep only those samples for which 3 out of the 5 models achieve less than 50% score (weighted) across the rubric criteria.

## A.2 DISTRIBUTION OF SAMPLES OVER SUBJECTS AND USE CASES

## A.3 RUBRIC CRITERIA TAG DEFINITIONS

As explained in Section 2.4, we annotate each rubric criterion with several tags to enable fine-grained analysis of model behavior. In Table 2, we provide definitions of the *evaluation dimensions* tag.

## A.4 GPT-5 STUDY MODE RESPONSE EXAMPLES

In Section 3.6, we provided details about why the study mode responses from GPT-5 collected from the website rank lower than GPT-5 itself. Here, we provide illustrative examples of cases where either the study mode response is indeed poorer or is truncated due to the model seeking user input before proceeding. In the latter case, the model response is incomplete (probably by design) and hence scores poorly against the rubric criteria. We present three examples in Figs. 8 to 10 where their responses are shown side-by-side along with the corresponding rubric criteria. These examples highlight some of the shortcomings of study mode: the responses are sometimes terse, or fail to get into the details of the problem, or sometimes expect a response from the student before proceeding. By grounding the comparison in rubric-based criteria, we provide concrete evidence for why study mode ranks lower in Table 1.

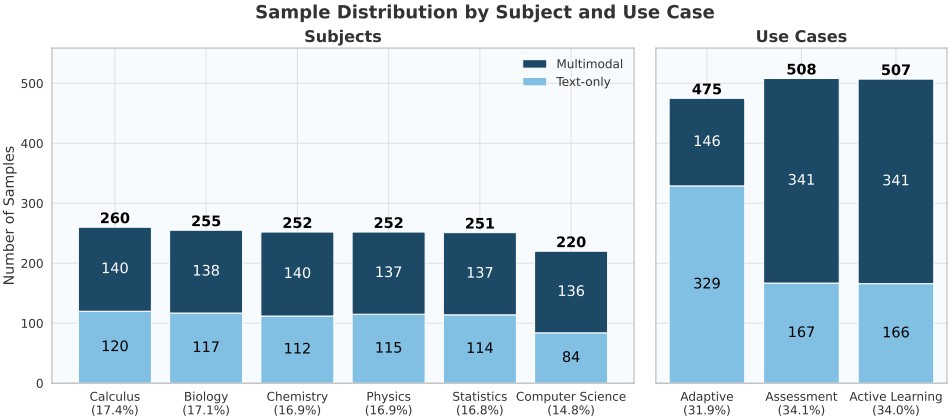

Figure 7: Distribution of samples by subject (left) and use case (right). The stacked bars indicate the number of Text-only (light blue) and Multimodal (dark blue) samples in each category. The distribution across the six subjects is relatively balanced, while the counts for the use cases vary more significantly. The labels 'Adaptive', 'Assessment', and 'Active Learning' correspond to adaptive explanation generation, assessment and feedback, and active learning support, respectively.

| Tag | Definition |
|---|---|
| Instruction following | Criterion checks for adherence to system instructions and prompt instructions |
| Truthfulness | Criterion checks for factual accuracy, capturing the "precision" of the model, and does not penalize missed content, incorrect formatting |
| Conciseness and Relevance | Criterion checks for how direct, on-topic, and efficient the content is. Examples are criteria that check for what should not be in the response, unnecessary formatting |
| Style and Tone | Criterion checks for clarity, fluency, and appropriateness of tone |
| Visual Perception [Multimodal Only] | Criterion checks whether the model correctly identifies the content from the image required to solve the problem. |
| Visual Reasoning [Multimodal Only] | Criterion checks whether the model correctly reasons based on the provided image. If the reasoning does not depend on the image, this tag is not used. |
| Student-level Calibration | Criterion checks if the model response accounts for the explicit/ implicit knowledge level of the student. |
| Emotional Component Calibration | Criterion checks if the model pays attention to a student's enthusiasm, frustrations, overwhelm, confusion, and other emotional cues mentioned in the prompt. |

Table 2: Definitions of the evaluation dimensions used to tag rubric criteria.

## A.5 DISTRIBUTION OF RUBRIC CRITERIA TAGS

In this section, we present the distribution of the four tags that each rubric criterion was annotated with when applicable. In Fig. 11, we show the distribution of the objectivity and explicitness tags. We observe that the majority of the rubric criteria are both objective and explicit. In Fig. 12a, we show the distribution of the evaluation dimensions, and in Fig. 12b, we show the distribution of tutoring skills across the rubrics.

## A.6 SYSTEM PROMPTS

To generate model responses for evaluation, we use system prompts to lightly guide the LLMs to assume the role of a tutor. We refrain from using overly prescriptive system prompts to measure the "natural" tutoring abilities of LLMs. This helps us create an evaluation system that is reflective of

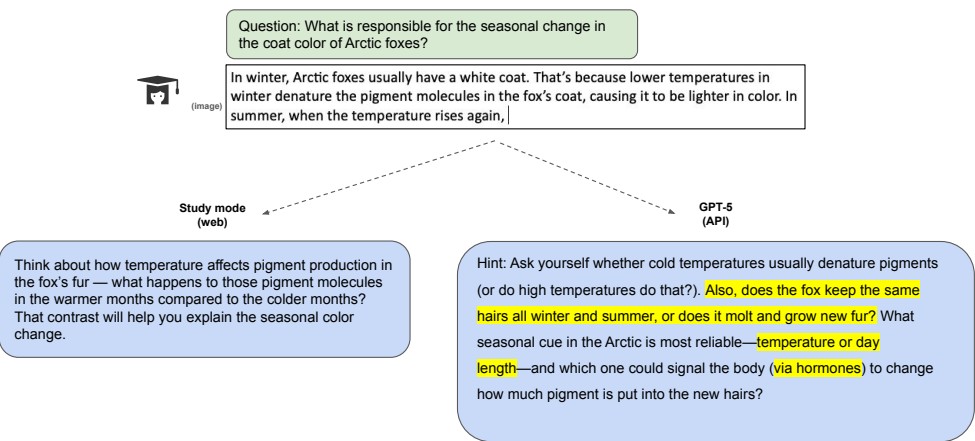

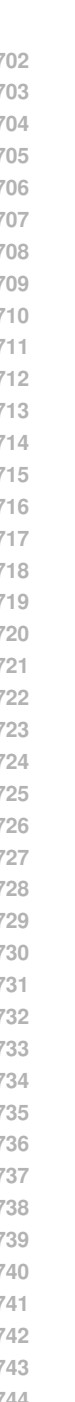

| | Study mode (web) | GPT-5 (API) |
|---|---|---|
| The response must acknowledge the student's current work by referring to their claim that lower temperatures in winter denature the pigment molecules in the fox's coat, causing it to be lighter in color. | 0 | 0 |
| The response must identify that the student is stuck due to a conceptual misunderstanding of what actually causes denaturation (i.e., high rather than low temperatures and extreme pH). | 0 | 1 |
| The response must hint that the seasonal coat-color changes observed in the Arctic fox cannot be caused by denaturation by encouraging the student to consider whether denaturation could occur at sub-zero arctic temperatures. | 0 | 1 |
| The response must hint that the seasonal coat-color changes observed in the Arctic fox cannot be caused by denaturation by encouraging the student to consider whether hair follicles whose pigment molecules are irreversibly denatured would ever be able to produce dark hairs again in the summer. | 0 | 0 |
| The response must provide a hint promoting the student to consider that denaturation is irreversible and therefore unlikely to explain the seasonal coat-color change. | 0 | 0 |
| The response must hint that a hormone affected by day length changes plays a role in regulating melanin production in Arctic foxes. | 0 | 1 |
| The response must hint that changes in day length can biologically influence fur color through melanin production. | 0 | 1 |
| The model must be able to reason that the student's response that the Arctic fox's pigment molecules are denatured in the winter is incorrect because denaturation cannot occur at the extreme low temperatures in which the Arctic fox lives. | 0 | 1 |
| The response should mention that the Arctic fox grows a new coat each season, which may help explain reversible changes in coat color. | 0 | 1 |
| The response must not state that the correct final answer is that the seasonal coat-color changes observed in the Arctic fox are due to photoperiodic melatonin regulation of melanin. | 1 | 1 |

Figure 8: An example comparing study mode and GPT-5 on the use case of *active learning support*. We can observe that the rubrics require the response to correct a misconception in the student regarding the relationship between temperature and fur color in arctic foxes and guide the student to consider other factors (such as daylight duration and hormonal changes) affecting the fur color during winter. As observed, the response from GPT-5 (API) addresses the rubric criteria directly, while the response from study mode is brief and does not refer to any other factors at all.

real-world usage, where students do not use elaborate instructions. Below, we outline the system prompt used for each use case.

**Adaptive explanation generation:**

- **Text-only samples:** You are an AI tutor helping a high school student understand a concept. Answer their question clearly and adjust your explanation based on what the student says they're confused about.

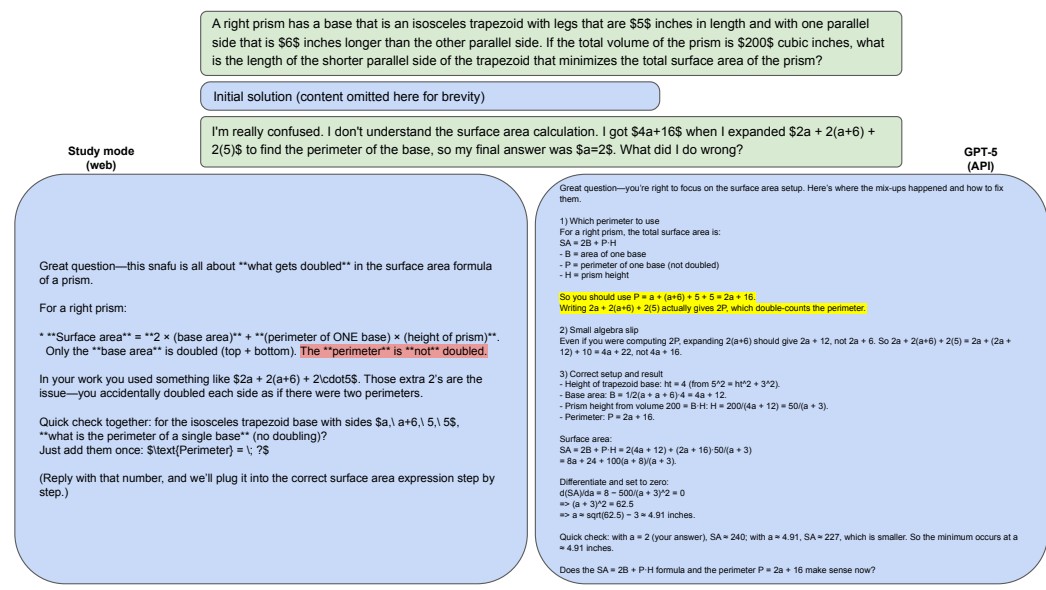

Figure 9: An example comparing study mode and GPT-5 on the use case of *adaptive explanation generation*. We can observe that the GPT-5 response clearly identifies that the student is confused about the formula of the perimeter and offers clarification, unlike study mode. Further, study mode does not provide the correct answer, but expects the student to respond before proceeding further.

- **Multimodal samples:** You are an AI tutor helping a high school student understand a concept. Answer their question clearly and adjust your explanation based on what the student says they're confused about.

**Assessment and feedback:**

- **Text-only samples:** You are an AI tutor reviewing a student's answer to a question. Evaluate whether it is correct, identify any mistakes, and explain your reasoning clearly. Provide an assessment of the student incorrect solution in the first response

- **Multimodal samples:** You are an AI tutor reviewing a student's answer to a question. Evaluate whether it is correct, identify any mistakes, and explain your reasoning clearly. Provide an assessment of the student incorrect solution present in the image.

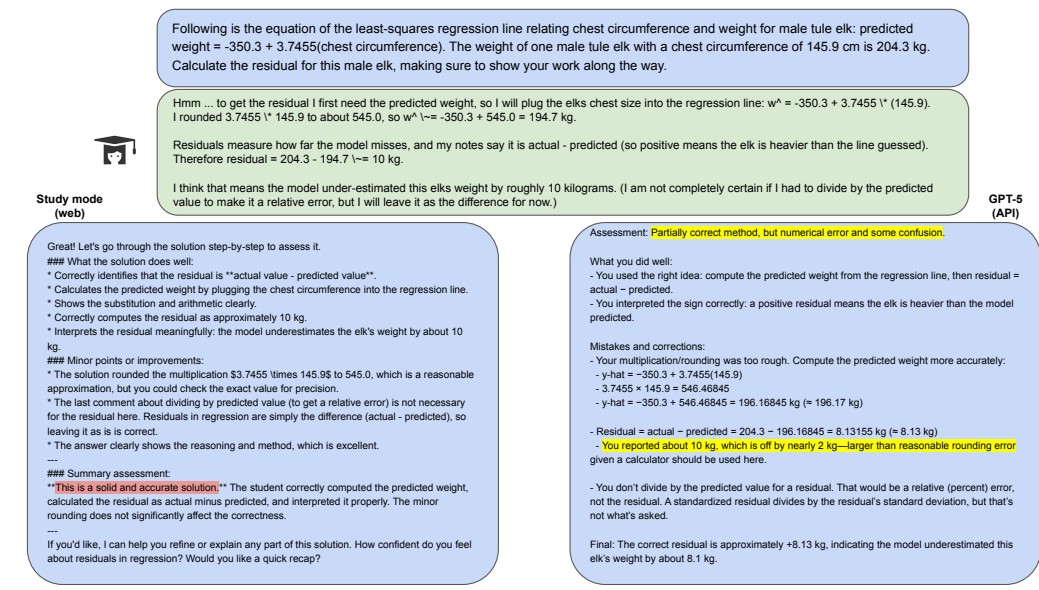

Figure 10: An example comparing study mode and GPT-5 on the use case of *assessment and feedback*. We can observe that study mode provides an inaccurate assessment and concludes that the student's solution was **solid**. However, GPT-5 identifies the right source of error and provides suggestions on how to avoid such errors in the future.

Note that for text-only samples, the initial solution appears in the conversation history as a model response to the initial question. Hence, the system prompt refers to it as the 'first response'.

**Active learning support:**

- **Text-only samples:** You are an AI tutor helping a student who got stuck partway through a problem. Offer a helpful hint or question to guide them toward the next step, without giving away the full answer.

- **Multimodal samples:** You are an AI tutor helping a student who got stuck partway through a problem. Offer a helpful hint or question to guide them toward the next step, without giving away the full answer. The image has the student partial solution you have to see in

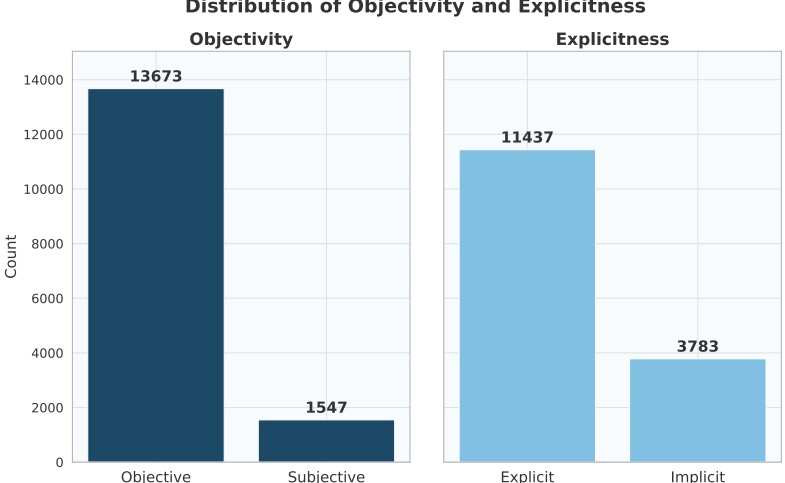

Figure 11: Distribution of objectivity and explicitness.

order to provide your helpful hints or questions to guide them toward the next step, without giving away the full answer

While manually collecting responses from the GPT-5 study model from its website, we use the following preambles:

**Adaptive explanation generation:**

- **Text-only samples** You will be presented with an initial question, a solution to the question, and a follow-up question seeking clarification. Help answer the follow-up question.

- **Multimodal samples:** You will be presented with an initial question, a solution to the question, and a follow-up question seeking clarification. The initial question will refer to the attached image. If an image is not provided, do not answer the question and say 'Image not provided'. Otherwise, answer the follow-up question.

**Assessment and feedback:**

- **Text-only samples:** You will be presented with an initial question and a solution to the question. Provide an assessment of the solution.

- **Multimodal samples:** You will be presented with an initial question and a solution to the question. Provide an assessment of the solution.

**Active learning support:**

- **Text-only samples:** Offer a helpful hint or question to guide me toward the next step.

- **Multimodal samples:** Offer a helpful hint or question to guide me toward the next step. My solution is shown in the attached image

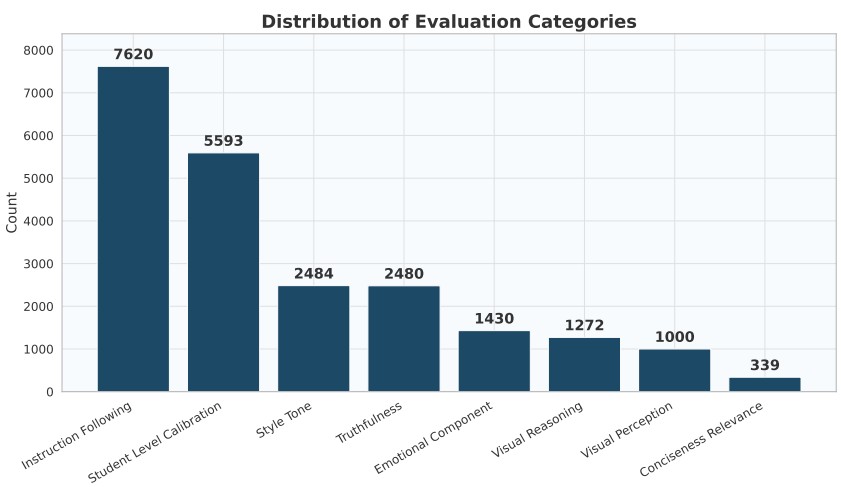

(a) Evaluation dimensions

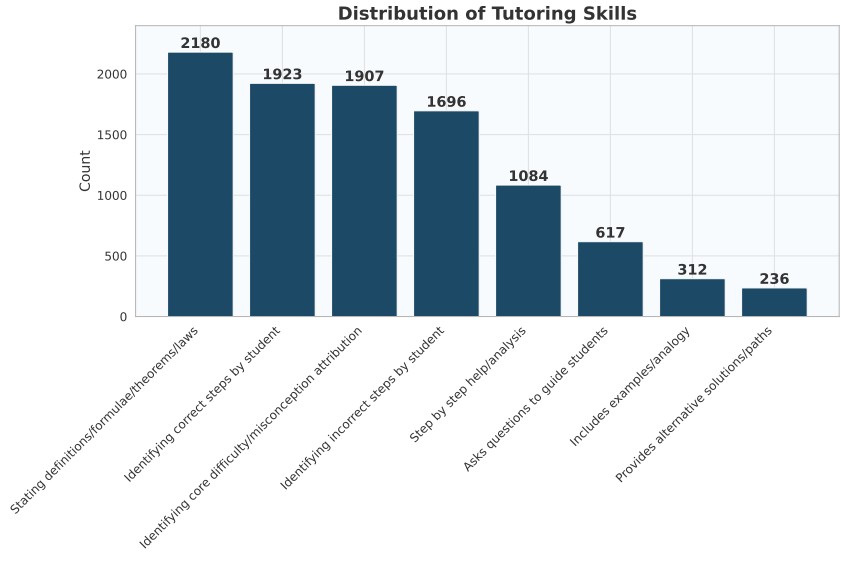

(b) Tutoring skills

Figure 12: Distributions across (a) evaluation dimensions and (b) tutoring skills

