# OpenReview forum: "TutorBench: A Benchmark To Assess Tutoring Capabilities Of Large Language Models"
_ICLR.cc/2026/Conference — ICLR 2026 Conference Withdrawn Submission_

### Official Review · Reviewer_qMyP · 2025-10-31

**Soundness:** 3
**Presentation:** 3
**Contribution:** 3
**Rating:** 6
**Confidence:** 3

**Summary:**

The paper introduces TUTORBENCH, a benchmark evaluating LLMs’ tutoring abilities beyond knowledge recall. The dataset contains 1,490 high-school/AP STEM samples spanning three tasks: adaptive explanation, assessment/feedback, and active-learning support. Each sample includes fine-grained, weighted, sample-specific rubrics for evaluation via an LLM judge. The benchmark includes multimodal inputs, with just over 800 samples containing real-world student-work images. A range of state-of-the-art models are evaluated; the best achieve only ~56% overall, indicating substantial headroom. Performance analyses by task, cognitive category, and tutoring skill reveal consistent weaknesses, particularly in personalisation and guided learning.

**Strengths:**

The paper addresses an important but underexplored dimension of LLM evaluation: tutoring proficiency. The benchmark spans six STEM subjects and three realistic tutoring scenarios. Its sample-specific, weighted rubrics enable structured, fine-grained evaluation of otherwise subjective tutoring behaviors. The dataset’s multimodal nature increases realism. Extensive analysis identifies performance trends across tasks, tutoring skills, and Bloom’s taxonomy. Validation of the judge against human experts (F 0.81) reinforces evaluation credibility. Results demonstrate a large performance gap between current models and desired tutor capabilities, underscoring the benchmark’s relevance.

**Weaknesses:**

The benchmark focuses on three narrow tutoring scenarios, excluding other educational tasks. Only final responses are evaluated, limiting assessment of multi-turn adaptivity.

Coverage is limited to STEM domains, excluding humanities, where tutoring differs significantly. Because rubrics are expert-written, stylistic bias is possible. The benchmark’s curated difficulty (removing “easy” samples) may over-emphasize failure cases and reduce representativeness.

**Questions:**

1. How consistent is the LLM-as-a-judge across models, and do some models appear to exploit rubric phrasing more than others?

2. What procedures ensured rubric construction consistency, and were inter-author agreement checks performed at design time?

---

### Official Review · Reviewer_Z3Ja · 2025-10-31

**Soundness:** 2
**Presentation:** 2
**Contribution:** 2
**Rating:** 2
**Confidence:** 4

**Summary:**

The paper introduces a benchmark for evaluating LLMs in tutoring settings, focusing on three tasks: (1) generating adaptive personalized explanations to student questions, (2) providing feedback on student's incorrect work, and (3) generating pedagogical hints for active learning. The authors construct the dataset using human experts in their respective fields, accompanied with sample-specific rubrics. To automate the evaluation, they develop an LLM-judge pipeline for scoring LLM-generated responses based on expert-written rubric items. Finally, the authors conduct an in-depth analysis of different models' performance on the new benchmark.

**Strengths:**

**Benchmark**

- A subset of the samples include images, which better aligns with real-world usage of the models in tutoring settings.
- Tutor and persona turns, as well as ground truth responses accompanied with rubric items are written by human experts.
- LLM-based automatic evaluation for more scalable benchmarking.

**Weaknesses:**

**Ground Truth**

In educational settings, it is often the case when there is no single ground truth response. Thus, constructing the benchmark on a single ground truth response inherently limits the evaluation of a broad range of possible pedagogically-appropriate responses. The proposes setup further exacerbates the issue as observing just a single turn from a student, different experts can make diverging conclusions about the student's knowledge state and thus employ different approaches for addressing the student's misconception(s).

First, to study the space of possible pedagogical responses for a given sample, multiple human experts can write a ground truth response and the corresponding rubrics. An analysis of agreement and coverage of human-written responses would help understand the problem better.

Second, to reduce ambiguity in inferring the student's misconceptions, one can generate more context about the student. A more realistic setting would be developing a student-specific context, including their prior questions, interactions, submissions, etc. and using as an input to the AI tutor.

**Study Mode Analysis**

The insights reported in Section 3.6 were surprising: OpenAI's special study mode performs worse than GPT-5 and Gemini-2.5-Pro. Can this be a limitation of the benchmark? One of the reported differences between GPT-5 and OpenAI's Study Mode is conciseness and not explicitly addressing the student's misconception (Figure 8). Is it possible that OpenAI's Study Mode is pedagogically more appropriate in these settings? If this is the case, then the benchmark can inadvertently penalize pedagogical responses and models.

**Limited Analysis**

The current analyses of the paper studies the performance of LLMs broken down by different category tags. However, an in-depth study of the failure modes accompanied with qualitative examples is missing.

Additionally, one can study the effect of the system / user prompt on the models' performance. Would the failure modes be mitigated by prompting? How much can prompting improve the current results (i.e., what's the actual ceiling of the benchmark)?

**Data Collection**

The paper misses important details about the data collection process. What was the medium of the data collection? What instructions were provided to the human experts?

**Lack of Grounding in Education Research**

The paper overlooks prior education research on human tutoring:
- What makes a tutor response pedagogically appropriate?
- How can we promote active learning in tutoring sessions (e.g., Socratic learning)?

**Questions:**

- What was the medium of the data collection? What instructions were provided to the human experts?

- How do the authors come up with the category tags for the rubrics? Are they grounded in prior work or do the authors propose a new taxonomy?

- For annotating the difficulty level in Section 3.4, is the ground truth expert-written response being annotated?

- What was the effect of the system prompt on the performance? How *"sensitive"* is the benchmark to the system / user prompt?

---

### Official Review · Reviewer_U6oS · 2025-11-01

**Soundness:** 1
**Presentation:** 2
**Contribution:** 2
**Rating:** 0
**Confidence:** 5

**Summary:**

TUTORBENCH is a benchmark for LLM tutoring for high-school and AP-level curricula. It focuses on three use cases: adaptive explanation, assessment/feedback, and active-learning support. The benchmark is multi-modal and uses LLM-as-a-judge for rubric creation and evaluation. The benchmark is multi-modal (which is great for an educational task) and is designed to be a "difficult" tutoring task for high school level courses and reports the quality of tutoring for 10 different models.

**Strengths:**

The benchmark is designed as a multi-modal setting where students' incorrect or incomplete answers (which are also partially generated by an LLM and not entirely real-world data) receive tutoring guidance and grading based on a rubric. Evaluating and testing tutoring behavior in LLMs is necessary and interesting. The benchmark is designed for high school curricula (though subjects are not listed in the paper) across multiple domains. The use of sample-level rubrics makes the evaluation more reproducible. The authors compare a wide set of models and provide per-use-case breakdowns. The paper uses judge–human agreement and attempts to quantify reliability, which is a step in the right direction.

**Weaknesses:**

The work introduces an educational benchmark and attempts to ground it also in the educational literature; however, there are some issues with the methodology and evaluation in the paper that I will list below:

The paper uses “active learning” incorrectly. Active learning is an instructional approach where students engage in activities (problem-solving, reflection, and active lecture viewing). The paper measures hint-based tutoring support that may promote active learning. The correct terminology is “hint-based scaffolding to support active learning”. This also needs to be reflected in the evaluation rubric. Currently, the evaluation rubric in the active learning use case is: "identifying core difficulty, identifying students’ correct steps, identifying students’ incorrect steps, recalling and stating knowledge, providing alternative solutions, including examples, asking questions to guide students, and providing step-by-step help." Although I have to admit that it is not very clear what rubric is used for what specific use case because each section introduces a new rubric and a new evaluation criteria; compare sections 3.5, 3.4, 3.3 (unclear how "recognizing student emotions such as confusion, frustration, curiosity, and generates responses with the right tone and style" is evaluated and measured), 3.1, and 2.4.

The boundary between adaptive explanation (use case 1) and active-learning support (use case 3) is unclear. Figure 1 and the examples do not make it obvious which goal is being evaluated or how annotators would consistently separate them. Provide paired, minimal contrasting examples of how each use case is seen in LLM outputs and how they're evaluated.

The LLM-as-judge section is thin for subjective criteria. Reporting variance vs. three experts is useful, but the paper does not show within-judge variance across seeds, across-judge variance (swap-judge model), or sensitivity to prompt wording. Figure 6 should also include repeated judge ratings for the same samples.

The dataset is described as expert-curated and reflective of real student use, yet conversations are LLM-generated and filtered by SOTA failures. This risks model-family bias and confounds claims of superiority.

The personalization claims are overstated for 1–3 turn interactions. With such short context, a tutor cannot infer misconceptions or learning characteristics. On that note, the misconception extraction setup (Figure 2) is not realistic. Real tutors do not receive a fixed list of misconceptions to choose from; a rubric-matching task is not the same as open-ended diagnosis. The paper links misconception extraction to “active learning” and then evaluates whether LLMs detect LLM-generated misconceptions, without validating the extraction itself against human-coded labels or established inventories. This is an area that the paper needs more grounding in educational litertaure to make the dataset and task more relevant to the education community or be helpful for the education community.

As mentioned earlier, rubrics and dimensions are introduced but not reported consistently. Terms like “explicit/implicit” and “objective/subjective” appear without analysis, while Figures 4–5 pivot to other LLM-generated dimensions. Bloom’s taxonomy is hierarchical; reporting flat category rates misses conditional mastery across levels and weakens the pedagogical grounding.

**Questions:**

Most of the issues and questions are listed in the earlier section, but to be more concrete, knowing the following would be helpful:

- What evidence supports personalization beyond immediate adaptation in a short exchange?
- Can you add side-by-side examples that differ only in instruction “explain” (use case 1) vs “hint without revealing the answer” (use case 3), and show distinct rubrics?
- Can you consolidate the rubric dimension set and add Bloom-conditional analyses?

**Details Of Ethics Concerns:**

I am not sure to what extent real student data is used and if so, it would be helpful to know whether the work has obtained an IRB approval.

---

### Official Review · Reviewer_D5GB · 2025-11-10

**Soundness:** 3
**Presentation:** 2
**Contribution:** 1
**Rating:** 4
**Confidence:** 3

**Summary:**

The authors provide a benchmark of approx. 1500 conversations across 6 STEM subjects of humans with a "tutor-persona". They evaluate 16 LLMs on this benchmark and find that Claude has best support. Evaluation itself is performed using LLMs that used rubrics; experiments with human are performed to highlist that LLM evaluation is accurate.

**Strengths:**

The contribution is timely as several commercial LLMs (by OpenAI, Anthropic, etc) have started to offer a "study mode". A good selection of dimensions regarding what makes a competent tutor is studied, and limitations are acknowledged.

**Weaknesses:**

This paper seems to be an incremental improvement over https://arxiv.org/abs/2402.11111  from 2024 which has approximately half as many samples (800+), and echoes several ideas from the current paper. What is called here a "rubric", seems to be called a "key point" in the linked paper.

Give the close overlap, the authors should make a clearer section in the related work, to show how the current paper distinguishes itself from prior literature. Otherwise, beyond the novelty aspect of evaluating the new study mode of LLMs, I am not sure that the approach in itself is sufficiently novel.

**Questions:**

N/A

---

### Note · Authors · 2025-11-25

**Comment:**

We would like to thank the reviewers and area chairs for their time and engagement with our submission. After consideration, we have chosen to withdraw the paper at this time. We appreciate the effort that went into the review process and look forward to sharing an updated version of this work in the future.

**Withdrawal Confirmation:**

I have read and agree with the venue's withdrawal policy on behalf of myself and my co-authors.